# REG3A: A Multifunctional Antioxidant Lectin at the Crossroads of Microbiota Regulation, Inflammation, and Cancer

**DOI:** 10.3390/cancers17142395

**Published:** 2025-07-19

**Authors:** Jamila Faivre, Hala Shalhoub, Tung Son Nguyen, Haishen Xie, Nicolas Moniaux

**Affiliations:** 1National Institute of Health and Medical Research INSERM, UMR 1193, Paris-Saclay University, Paul-Brousse University Hospital, Hepatobiliary Centre, 94800 Villejuif, France; hala.shalhoub@inserm.fr (H.S.); tung-son.nguyen@inserm.fr (T.S.N.); haishen.xie@inserm.fr (H.X.); 2Medical School, Paris-Saclay University, 94270 Le Kremlin-Bicêtre, France; 3Assistance Publique-Hôpitaux de Paris (AP-HP), Université Paris-Saclay, Hospital-University Department of Biology and Genetics, Paul-Brousse Hospital, 94800 Villejuif, France

**Keywords:** REG3A, C-type lectin, oxidative stress, reactive oxygen species (ROS) scavenger, host-microbe interactions, inflammation, tumor progression, tumor suppression, immunomodulation

## Abstract

REG3A, a key member of the human REG lectin family, plays a multifaceted role in immunity, inflammation, and cancer. Primarily expressed in gastrointestinal epithelial cells, it reinforces gut barrier integrity, preserves mucosal immunity, and regulates host–microbiota interactions. Beyond its antimicrobial functions, REG3A acts as a targeted non-enzymatic antioxidant, protecting tissues from oxidative damage. Its expression is tightly regulated by inflammatory signals and is markedly upregulated during immune activation, where it limits microbial invasion, reduces tissue injury, and promotes repair. While REG3A offers critical protection in inflammatory settings, its role in cancer is far more complex. Depending on the tumor type and microenvironment, REG3A can either support tumor growth or exert tumor-suppressive effects. This review highlights the multifaceted biology of REG3A, with a focus on its roles in epithelial defense, immune modulation, oxidative stress regulation, and its paradoxical functions in cancer.

## 1. Introduction

The REG3 proteins, members of the regenerating islet-derived (REG) lectin family, were first identified in 1984 following their isolation from rat pancreatic islets [1]. Among them, REG3A has garnered particular interest due to its multifunctional roles in tissue regeneration, immune regulation, and host defense. Over time, REG3A has been designated by several names depending on the species and physiological conditions in which it was studied. It was initially described as Pancreatitis-Associated Protein (PAP or PAP1) after its identification in a rat model of acute pancreatitis [2], and as Hepatocarcinoma-Intestine-Pancreas protein (HIP) in human hepatocellular carcinoma [3]. Other less commonly used aliases have included peptide 23, Islet Neogenesis-Associated Protein (INGAP), and Pancreatic Thread Protein (PTP). While the term HIP/PAP persisted in the early literature, the protein is now widely referred to as REG3A, in line with its classification within the REG family of C-type lectins with regenerative properties. Notably, some studies have incorrectly referred to REG3A as REG2, which is in fact a distinct murine-specific family member without a human homolog [4].

The REG protein family is subdivided into four groups: REG1, REG2, REG3, and REG4. Except for REG4, which is located on chromosome 1, all other REG genes in humans are clustered on chromosome 2p12 (corresponding to chromosome 16p12–13 in mice) [5], reflecting their derivation from a common ancestral gene and subsequent duplication during evolution. This genetic clustering implies both functional conservation and potential redundancy among family members. The REG1 subgroup comprises REG1A and REG1B in human [6,7] and mice, while the REG3 subgroup includes REG3A and REG3G in humans, and Reg3α, Reg3β, Reg3δ, and Reg3γ in mice. All REG3 members share significant sequence homology both within and across species [5]. Human REG3A shares 69.1% and 66.3% sequence identity—and 80.9% and 82.4% similarity—with mouse Reg3β and Reg3γ, respectively. All three proteins conserve the functional EPN lectin motif at the apex of loop 1. In loop 2, however, the ERN motif found in REG3A and Reg3β is substituted by an ETN motif in Reg3γ. Cysteine and aromatic residues are fully conserved across these orthologs. The primary sequence divergence lies in the carboxy-terminal region, for which no defined functional motif has been identified. Mouse Reg3α shows greater divergence, featuring a QPN motif in loop 1 and a DGD sequence in loop 2, indicating distinct lectin-binding specificity. Among these, Reg3β and Reg3γ are considered the murine orthologs of human REG3A, while Reg3α is orthologous to human REG3G. The diversity of nomenclature across species and the frequent use of different orthologs in experimental models have complicated the functional characterization of REG3A. For the sake of consistency, the term REG3A will be used throughout this article to refer to both the human protein and its murine counterparts Reg3β and Reg3γ.

REG3A is initially synthesized as a 175-amino acid prepropeptide (19 kDa) (Figure 1) [3,8,9]. Proteolytic removal of the 26-amino acid signal peptide results in the secretion of the mature ~16 kDa form into the intestinal lumen and, to a lesser extent, into the systemic circulation [9,10]. The N-terminal pro-segment keeps REG3A in an inactive state until cleaved by host or microbial proteases. Activation occurs through cleavage at specific sites by digestive enzymes, trypsin at Arg37–Ile38 and elastase at Ser35–Ala36, or by bacterial proteases such as *Bacillus subtilis* npr9 [11,12,13]. In vitro studies have shown that 1.5 µM REG3A is fully cleaved by 10 nM trypsin within 5 min, and by 50 nM and 150 nM of elastase 2A and 2B, respectively, within 30 min. Similarly, *Bacillus subtilis* culture supernatant can fully cleave 1.5 µM REG3A within the same time frame [13]. The bacterium-aggregating activities of the resulting cleavage products were evaluated and found to be comparable. Interestingly, trypsin cleavage generates a 15 kDa polymerizable form that assembles into insoluble, fibrillar structures, whereas elastase cleavage preserves REG3A in a soluble form [12,14].

The three-dimensional structure of REG3A has been resolved by X-ray crystallography (PDB ID: 1UV0 [10] and NMR spectroscopy (PDB ID: 2GO0) [11,15]). Its structure includes two α-helices, eight β-sheets, and three disulfide bridges (Cys 40-Cys51, Cys68-Cys171, Cys146-Cys163), forming a classical carbohydrate recognition domain (CRD) (Figure 1). REG3A belongs to subgroup VIII of C-type lectins, which are characterized by a single CRD that binds carbohydrates in a calcium-independent manner, typically through conserved motifs such as EPN (Glu–Pro–Asn), QPD (Gln–Pro–Asp), or QPN (Gln–Pro–Asn) [8,16,17]. REG3A uniquely positions its EPN motif on loop 1 (residues 107–121) and the ERN motif on loop 2 (residues 130–145), contributing to its distinct ligand specificity [18] (Figure 1).

Functionally, REG3A is well known for its lectin activity, particularly its ability to bind pathogen-associated molecular patterns (PAMPs) such as mannose and N-acetylglucosamine (GlcNAc) residues [19]. Its affinity increases significantly when these carbohydrates are part of polysaccharide structures like peptidoglycan, lipopolysaccharides (LPS), mannan, and chitin. This binding underlies its potent bactericidal activity against Gram-positive microorganisms, such as *Enterococcus faecalis* and *Listeria* spp., especially in its cleaved, active form within the intestinal lumen [19]. In addition to microbial recognition, REG3A can bind glucose and its phosphorylated derivatives involved in the hexosamine biosynthesis pathway, thereby influencing key metabolic processes such as protein O-GlcNAcylation, complex glycosylation, and glycolysis [20].

Beyond its lectin functions, REG3A has emerged as a potent reactive oxygen species (ROS) scavenger, with a strong affinity for hydroxyl radicals [21]. Accumulating evidence supports its role in protecting tissues against oxidative injury. In acute liver failure, REG3A binds to fibrin scaffolds in necrotic areas of the liver, where it reduces ROS levels and promotes hepatocyte survival [22]. In muscle cells, REG3A prevents oxidative damage to the glycoprotein GP130, thereby preserving GP130/AMPK signaling and enhancing glucose uptake [23]. However, the underlying antioxidant mechanisms remain incompletely understood, particularly whether REG3A engages with canonical redox pathways such as Nrf2 signaling. Given the pivotal role of Nrf2 in regulating antioxidant responses, it is plausible that REG3A may influence or interact with this pathway. Further studies are warranted to determine whether REG3A modulates Nrf2 activity, either directly or indirectly through its effect on oxidative stress, as well as its potential impact on other antioxidant mediators, including superoxide dismutase (SOD), catalase, and heme oxygenase-1 (HO-1).

In summary, REG3A exhibits a multifunctional profile combining lectin activity, antimicrobial properties, and ROS-scavenging capacity. It binds bacterial cell wall carbohydrates in a calcium-independent manner, disrupts bacterial membranes to exert bactericidal effects, or protects oxygen-sensitive commensals from oxidative damage. These diverse functions contribute to epithelial barrier defense, immune modulation, and the maintenance of tissue homeostasis. The following sections will examine the implications of REG3A’s dual functionality in inflammation, host–microbe interactions, and tumor biology.

## 2. REG3A: Expression and Regulation

REG3A is predominantly expressed in gastrointestinal organs such as the pancreas [24], stomach [25], and intestinal tract [26]. Its expression is highly inducible in response to infection and inflammation (Table 1). At lower levels, REG3A is also detected in a limited number of other cells and tissues, including cholangiocytes [27], skin [28], uterus [29], and the pituitary gland [30]. A key driver of REG3A expression in the gut is bacterial colonization [19,31]. In germ-free or antibiotic-treated animals, REG3A expression in the intestinal lumen is minimal, whereas probiotic supplementation significantly enhances its expression [19,31,32,33]. Paneth cells play a central role by sensing bacterial signals through the TLR-MyD88 signaling pathway, thereby promoting REG3A transcription [34]. In addition, IL-22, produced by innate lymphoid cells, further stimulates REG3A expression [35], thereby enhancing its role in maintaining intestinal barrier function and in protecting intestinal stem cells. REG3A plays a role in shaping the gut microbiome and maintaining spatial separation between bacteria and the intestinal epithelium [36].

REG3A expression is modulated under various pathophysiological conditions. It is robustly upregulated during bacterial infections, supporting its function as a protective factor against microbial invasion [37,38]. For instance, infections with *Listeria monocytogenes* and *Salmonella enteritidis* induce REG3A expression in the intestine, with corresponding effects on mucus distribution [39]. Furthermore, the promoter region of the REG3A gene contains a functional IL-6 responsive element, highlighting a close connection between REG3A regulation and the immune response [40]. Other cytokines including IL-8, IL-17, IL-22, IL-33, and IL-36, have also been shown to upregulate REG3A expression [41,42,43,44]. Overall, REG3A functions as an acute-phase protein and is a key effector in the innate immune response.

REG3A expression is markedly upregulated in response to both acute and chronic tissue stress, as well as across a range of malignancies. A comprehensive summary of these expression patterns is provided in Table 1 and Table 2. Owing to its strong diagnostic relevance, REG3A has gained recognition as a circulating biomarker in several pathological conditions, where elevated serum levels often correlate with disease severity and poor prognosis. In chronic colitis [45], gastrointestinal graft-versus-host disease (GI-GvHD) [46], sepsis [47], and pancreatic ductal adenocarcinoma (PDAC) [48], increased REG3A levels reflect heightened disease activity. Notably, in GI-GvHD, plasma REG3A concentrations are significantly higher in affected patients compared to those without gastrointestinal involvement. A threshold of ≥72 ng/mL has been shown to predict a 1.9-fold increased risk of non-relapse mortality, independent of other clinical variables, establishing REG3A as a validated prognostic marker in this setting [46,49]. In sepsis, REG3A has recently emerged as both a diagnostic and prognostic biomarker. Serum levels correlate strongly with clinical severity scores (SOFA and APACHE II), accurately distinguish septic patients from healthy controls, and independently predict 28-day survival [47]. Similarly, in rectal cancer patients undergoing neoadjuvant chemoradiotherapy (CCRT), transcriptomic analyses identified REG3A as the most upregulated gene in non-responders (log_2_ ≈ 1.25, *p* = 0.0079). High REG3A protein expression is associated with poor treatment response, increased lymph node metastasis, vascular invasion, and serves as an independent predictor of both disease-specific and metastasis-free survival [50]. Furthermore, REG3A shows promising utility as a urinary biomarker in bladder cancer. In a single-cohort study, REG3A demonstrated a sensitivity of 80.2%, specificity of 78.2%, positive predictive value of 75.7%, negative predictive value of 82.3%, and an area under the curve (AUC) of 0.863, outperforming conventional markers such as NMP22 (sensitivity 52.1%, specificity 93.5%) [51]. Collectively, these findings underscore the emerging clinical value of REG3A as a biomarker in both inflammatory and oncologic gastrointestinal diseases.

**Table 1 cancers-17-02395-t001:** REG3A expression patterns across tissues and clinical conditions.

Organs	Condition		
	Physiological	Inflammatory	Malignant
**Pancreas**	β-cells (islets)	Pancreatitis [24], (inflamed acini) Cystic fibrosis [52]	Adenocarcinoma [53]
**Stomach**	Epithelial cells	N/A	Gastric cancer [54]
**Small intestine**	Paneth cells	Paneth cells Enterocytes	
**Colon**	Not expressed	Colitis [55], Celiac disease, Crohn’s disease [56] (enterocytes) [55]	Colorectal cancer [57]
**Liver**	Cholangiocytes	Hepatitis [21] (inflamed hepatocytes) [58,59]	Hepatocellular carcinoma [3,60] Cholangiocarcinoma [27]
**Bladder**	Not expressed	N/A	Bladder cancer
**Skin**	Keratinocytes	Skin lesions Psoriasis [28]	
**Mammary gland**	N/A	N/A	Breast cancer
**Brain**	Motoneurons [61]	Brain injury [62] Alzheimer [63]	
**Pituitary gland**	Growth hormone producing cells	N/A	Adenoma [64]
**Lung**	N/A	Fibrosis [65]	Lung cancer
**Trachea**	N/A	N/A	Squamous cell carcinoma [66]
**PBMCs**	N/A	Lupus nephritis [67], Polymyositis Dermatomyositis [68]	Gastric cancer [54]

Abbreviations: N/A: not available.

**Table 2 cancers-17-02395-t002:** Comparative analysis of REG3A studies across cancer types and experimental models (in vitro, in vivo, human cohorts).

	Link With Cancer	Conditions		
		In Vitro	In Vivo	Human Cohort
**Colorectal** **cancer**	**Suppressor**	BMI1/MEL18 repress REG3A transcription [69] REG3A binds IL6/GP130 [69] REG3A inhibits IL6/STAT3 signaling [69]	rREG3A reduces polyps in AOM-DSS model [69]	REG3A inversely correlates with cancer (331 pairs NT/T) [57] High REG3A associates with better OS (*n* = 279) [57,69] REG3A inversely correlates with venous invasion [57] REG3A inversely correlates with STAT3 activation [69]
	**Promoter**	REG3A induces AKT and ERK [70]	REG3A promotes tumor growth in xenografts [70]	REG3A positively correlates with cancer (*n* = 335) [70] High REG3A associates with tumor size, differentiation, stage (*n* = 82) [70]
**Gastric** **cancer**	**Suppressor**	REG3A inhibits AKT and GSK3β activation. REG3A induces apoptosis. REG3A reduces cell invasion [71]	N/A	REG3A negatively correlates with cancer (30 NT/T pairs [25]; *n* = 876 [71]) High REG3A associates with better OS (*n* = 311) and disease-free survival (*n* = 565) [71]
	**Promoter**	REG3A inhibits proliferation, migration and invasion [72] REG3A suppresses JAK2/STAT3 signaling [72]	N/A	REG3A upregulated in cancer (41 NT/T pairs) [72]
**Hepato-** **cellular** **carcinoma**	**Suppressor**	Wnt/β-catenin induces REG3A [60] REG3A reduces proliferation [20] REG3A binds hexosamine pathway sugars [20] REG3A lowers O-GlcNAcylation [20]	REG3A inhibits cancer development (WHC/MYC, DEN) [20] REG3A lowers UDP-GlcNAc [20]	REG3A localizes to cirrhotic hepatocytes [26,27,58] REG3A links to low stage HCC, no vascular invasion, β-catenin mutation (*n* = 265) [73] REG3A associates with β-catenin activation (42 NT/T) [60] REG3A improves disease-free survival in cirrhosis (*n* = 216) [20] No link between REG3A and OS in patients with HCC (*n* = 507) [20]
	**Promoter**	HSC/HCC coculture induces REG3A via PDGFββ, reducing apoptosis [74] REG3A knockdown in coculture lowers p42//44 activation [74]	REG3A reduces tumor growth of Lx2/MH134 xenografts in C3H mice [74]	REG3A positively correlates with cancer (88 NT/T pairs) [74]
**Pancreatic cancer**	**Suppressor**	N/A	N/A	N/A
	**Promoter**	REG3A promotes proliferation, migration, and invasion; activates EGFR and GP130/JAK2/STAT3 [48,75,76,77] IL6 induces REG3A via JAK2/STAT3 [75] REG3A inhibits DC maturation [78] REG3A drives acini-to-ductal metaplasia via EXTL3/RAS/ERK (3D culture) [79]	REG3A promotes cancer in DMBA-caerulein and Pdx1-Cre; Kras^G12D^; Ink4a/Arf^fl/fl^ models [48,76,79,80] REG3A enhances tumor growth and induces immunosuppression in Panc02 xenografts [78] REG3A activates EGFR/JAK2/STAT3 and synergizes with IL6 to boost tumors in xenograft mice [75,78]	Serum REG3A < 17.5 µg/L links to cancer (*n* = 254) [48], high TNM grade (*n* = 58) [77] and poor survival [48,77] REG3A localizes to peritumoral acini and CK19^+^ preneoplastic lesions [48,77,79] REG3A correlates with IL6/JAK/STAT3 in peritumoral acini [77]

Abbreviations: rREG3A: recombinant REG3A; OS: overall survival; HCC: hepatocellular carcinoma; HSC: hepatic stellate cell; DC: dendritic cell; N/A: not available.

## 3. REG3A: Balancing Defense and Harmony in the Gut

REG3A plays a central role in intestinal antimicrobial defense, with a preference for targeting Gram-positive bacteria. Cash et al. showed that its luminal, truncated form exerts potent bactericidal activity against organisms such as *Enterococcus faecalis* and various *Listeria* strains [19]. This effect is mediated through the direct binding of REG3A to bacterial peptidoglycan, which anchors the protein to the bacterial surface. Upon binding, REG3A facilitates pore formation in the membrane, leading to cytoplasmic leakage and bacterial cell death [11,81] (Figure 2). Through this mechanism, REG3A functions as an antimicrobial protein (AMP), cooperating with other intestinal AMPs to reinforce the epithelial barrier and maintain gut homeostasis [82]. Interestingly, REG3A also demonstrates bactericidal activity against select Gram-negative bacteria, including *Clostridium butyricum*, *Lactobacillus reuteri*, and several *Escherichia coli* strains [83]. This activity is thought to be mediated via binding to LPS, the hallmark component of Gram-negative outer membranes [84] (Figure 2). By exerting these antimicrobial effects, REG3A contributes to maintain spatial segregation between gut microbes and the epithelium surface [36], a critical defense that limits bacterial translocation to extra-intestinal sites, such as the liver [85].

Although a number of Gram-positive and Gram-negative species have been classified as either susceptible or resistant to REG3A, the molecular basis for this selectivity remains incompletely understood. Studies using *Salmonella Typhimurium* mutants with modified LPS structures suggest that REG3A’s activity may depend on its ability to recognize specific peptidoglycan and LPS configurations [83]. Understanding these interactions is further complicated by interspecies differences: whereas humans express a single REG3A isoform, mice produce two orthologs, REG3α and REG3γ, which may differ in their bacterial specificity. The role of REG3A in host–microbe interactions extends far beyond pathogen clearance (Figure 2). Rather than indiscriminately clearing bacteria, a strategy that could be evolutionarily detrimental, REG3A promotes intestinal immune homeostasis by sculpting a balanced and protective microbial ecosystem. Compelling evidence revealed that REG3A shapes gut microbiota composition, enriching Clostridiales, while depleting *Bacteroidetes* and *Proteobacteria*. This microbial shift correlated with reduced oxidative stress and intestinal inflammation, leading to a lower susceptibility to colitis (Figure 2). Notably, REG3A was shown to decrease ROS levels in the gut microbiota of mice with colitis and enhance the survival of oxygen-sensitive, Gram-positive *Clostridia* in vitro, suggesting that its antioxidant activity contributes to microbiota modulation [86].

Animal studies highlight the diet-sensitive nature of REG3A expression and its influence on gut microbial ecology. Frazier et al. [87] showed that REG3A expression in the intestinal epithelium follows a diurnal rhythm, paralleling oscillations in *Lactobacillus* abundance and suppression of *Peptostreptococcaceae*. A high-fat diet disrupts this pattern, flattening REG3A expression, reducing *Lactobacillus* fluctuations, and promoting *Clostridiaceae* and *Peptostreptococcaceae* expansion, microbial shifts linked to impaired glucose regulation and metabolic dysfunction. In contrast, fermentable fibers such as inulin markedly upregulate REG3A expression. Shin et al. [33] demonstrated that this fiber-induced REG3A expression is crucial for the metabolic benefits of dietary fiber, including improved glucose tolerance and the expansion of beneficial microbes like Bifidobacteria. Together, these findings establish REG3A as a key mediator linking diet, microbial dynamics, and host metabolic health.

## 4. REG3A in Innate Immunity: Sensing DAMPs and Modulating Anti-Inflammatory Responses

While REG3A is well-known for recognizing microbial sugars, it also binds to damage-associated molecular patterns (DAMPs), particularly within the inflammatory extracellular matrix (ECM) during acute hepatitis. In these settings, REG3A accumulates at sites of liver injury, localizing specifically to fibrin deposits formed through fibrinogen polymerization, a process triggered by hepatocyte cytolysis. Through its potent free radical scavenging activity, REG3A plays a pivotal role in mitigating local oxidative stress, thereby supporting hepatocyte survival and promoting liver tissue repair and regeneration. [21,22,88].

Under inflammatory conditions where hepatocyte death is prominent, REG3A exerts a potent cytoprotective effect [89]. In vivo, this protection is recapitulated in REG3A transgenic mice and engineered to express human REG3A under the liver-specific albumin promoter. These mice show improved survival in models of acute liver failure induced by acetaminophen or Fas receptor activation [90,91]. These results led to phase 1 and 2 trials of recombinant REG3A for the treatment of acute liver failure [88].

Beyond the liver, REG3A exhibits broad cytoprotective activity across various tissues. In the central nervous system, REG3A safeguards cerebellar, cortical, and hippocampal neurons from H_2_O_2_-induced oxidative damage, enhances neuronal plasticity [92,93], and reduces NMDA receptor-mediated excitotoxicity caused by ibotenate [92]. Moreover, REG3A acts as a chemoattractant, recruiting macrophages to sites of neuronal injury, thus supporting repair and regeneration [94]. Similar regenerative effects are observed in cardiac tissue post-myocardial infarction. There, REG3A expression is induced by oncostatin M, an IL-6 family cytokine that drives cardiomyocyte dedifferentiation and remodeling. REG3A promotes macrophage recruitment to protect the cardiac ECM from neutrophil-mediated degradation within the infarcted region, facilitating tissue repair [95,96].

Notably, REG3A also possesses anti-fibrotic properties in both liver and lung tissues. In mouse models, REG3A reduces fibrosis following CCl_4_-induced liver injury or bile duct ligation [58], as well as bleomycin-induced lung fibrosis [65]. REG3A attenuates oxidative stress and modulates stellate cell responses to TGF-β, thereby preventing excessive epithelial–mesenchymal transition and limiting fibrotic progression [58,65].

## 5. REG3A in Tissue Regeneration: A Critical Link Between Inflammation and Repair

As previously discussed, the localized recruitment and accumulation of REG3A at sites of tissue injury enables a targeted reduction in oxidative stress and supports tissue repair. The antioxidant function of REG3A underpins its potent pro-survival activity, playing a central role in tissue regeneration across various tissues. In the nervous system, REG3A promotes motor neuron regeneration and exerts neurotrophic effects by enhancing motor neuron survival [93]. It also stimulates Schwann cell proliferation, a crucial step in peripheral nerve repair [61]. Mechanistically, REG3A activates the Akt signaling pathway, contributing to the downstream effects of ciliary neurotrophic factor (CNTF), which is essential for neuronal survival [93]. In liver tissue, REG3A enhances epidermal growth factor (EGF)-induced hepatocyte proliferation, facilitating liver regeneration following partial hepatectomy [89,90,91]. This ability to support healing makes it a promising treatment for liver injury and recovery after surgery. In the skin, REG3A plays a dual role in regeneration and immune modulation. Lai et al. demonstrated that REG3A is expressed in keratinocytes of both the epidermis and hair follicles in psoriasis patients and mouse models. It promotes keratinocyte proliferation while inhibiting terminal differentiation, processes essential for wound re-epithelialization and the hyperproliferative phenotype observed in psoriasis. This mitogenic effect is mediated via the EXTL3-Akt-STAT3 signaling axis [28]. Moreover, REG3A contributes to immune homeostasis in the skin by inducing SHP-1, a negative regulator of Toll-like receptor 3 (TLR-3), thereby suppressing excessive inflammatory responses following skin injury. This regulation is particularly important in diabetic skin inflammation, where hyperglycemia impairs IL-33-dependent REG3A expression, leading a delayed wound healing [43].

Exostosin-like glycosyltransferase 3 (EXTL3), primarily known for regulating heparan sulfate biosynthesis [97], has been proposed as a putative membrane receptor for REG3A. Initially characterized as a receptor for REG1A [98], subsequent bioinformatics analyses and functional studies suggest that EXTL3 also mediates REG3A activity [99]. Binding to EXTL3 activates the RAS-RAF-MEK-ERK signaling cascade in tissues such as the pancreas and skin, supporting cell survival and proliferation [79]. Other receptors have also been implicated in REG3A signaling. These include receptors containing the gp130 subunit and the EGFR, both of which can activate the JAK2/STAT3 pathway, further highlighting the pleiotropic regenerative and immunomodulatory roles of REG3A [75,76].

## 6. Systemic Roles of REG3A: Linking Metabolism to Health and Disease

Since the discovery of INGAP, the hamster homolog of REG3A, interest has grown in the role of REG family proteins in glucose homeostasis and metabolic regulation. REG3A, in particular, has emerged as a critical factor in pancreatic regeneration and β-cell protection, positioning it as a potential therapeutic target in diabetes and metabolic disorders. Evidence from animal models supports this role. REG3A promotes β-cell survival and may contribute to islet neogenesis via mechanisms involving PI3K/ATF-2/Cyclin D1 signaling, β-cell proliferation, or progenitor cell differentiation [100,101,102,103,104]. Notably, administration of INGAPpp, a pentadecapeptide derived from hamster REG3A, has been shown to increase islet volume and cell mass in diabetic models, preventing streptozotocin (STZ)-induced diabetes without directly stimulating cell proliferation [99,105]. These findings prompted clinical trials in 2009, in which INGAP treatment increased C-peptide levels in type 1 diabetes and lowered HbA1c in type 2 diabetes. However, it did not significantly reduce blood glucose or insulin dependency [106]. Further studies by Xiong et al. confirmed that REG3A does not act as a mitogen for β-cells. In mice expressing REG3A under the RIP-1 promoter, pancreatic islets displayed normal histology and insulin secretion, yet were resistant to STZ-induced hyperglycemia and weight loss [107]. These results reinforce REG3A’s cytoprotective, rather than proliferative, role in β-cell maintenance.

In alcohol-induced metabolic dysfunction, REG3A plays a protective role in maintaining gut–liver axis integrity. Chronic alcohol intake downregulates murine REG3A homologs (Reg3β and Reg3γ), disrupting the spatial segregation between commensal bacteria and the mucosal surface. This leads to bacterial translocation, liver inflammation, and the progression from steatosis to steatohepatitis [59]. REG3A overexpression restores microbial balance and barrier function, attenuating liver injury [59,85]. Similarly, REG3A expression can be induced by prebiotics (e.g., fructo-oligosaccharides), microbial metabolites (e.g., indole-3-acetic acid), or IL-22-producing *Lactobacillus reuteri*, all of which protect against alcohol-related liver disease [44,59].

Beyond alcohol-related pathology, REG3A also shapes the gut microbiota in obesity and metabolic syndrome. REG3A promotes a *Lactobacillus*-rich intestinal environment, including strains like *Lactobacillus* NK318.1, which enhance anti-inflammatory macrophage populations in the gut. These macrophages migrate to adipose tissue, contributing to improved resistance to diet-induced obesity [108]. In a study by Shin et al., a single injection of REG3A preserved gut barrier function, maintained energy balance, and improved glucose regulation in mice on a high-fat diet, suggesting a potential endocrine-like role for REG3A in systemic glucose tolerance [33].

Intriguingly, both alcohol and high-fat diets suppress IL-22 and REG3A expression in the intestine [109]. Yet, intestinal inactivation of murine REG3A homologs does not always correlate with worsened liver outcomes; while bacterial translocation and steatohepatitis were unaffected in one study, endotoxemia was significantly increased following REG3A silencing [110], underscoring a complex and variable role in gut–liver homeostasis.

Finally, REG3A directly contributes to metabolic regulation beyond the gut and pancreas. In liver-specific REG3A transgenic mice, both lean and obese animals showed improved glycemic control and enhanced insulin sensitivity [23]. Similarly, recombinant REG3A administration in obese (ob/ob) or high-fat-diet-fed mice enhanced glucose uptake in skeletal muscle and activated AMPK in a GP130-dependent manner, while reducing protein oxidative damage [23]. These systemic effects support the potential of REG3A as a metabolic modulator affecting multiple organs.

## 7. REG3A and Cancer: A Dual-Edged Role

REG3A is a critical mediator of cellular survival and growth, with well-established roles in regulating proliferation, differentiation, immune responses, metabolism, and tissue regeneration. These functions are essential for preserving tissue integrity and homeostasis and have consequently drawn attention to REG3A’s potential involvement in cancer biology. Elevated REG3A expression has been reported in certain human tumor types and has been associated with cancer initiation and progression in various experimental models [27,53,57,70,72,74,111,112]. Nevertheless, its exact contribution to tumorigenesis remains to be elucidated, as the literature presents conflicting evidence regarding its pro- and anti-carcinogenic effects, with no clear understanding of phenotypic variations (Table 2).

Members of the REG3 family, including REG3A, have been extensively studied in inflammatory disorders. Their expression is frequently dysregulated in conditions such as pancreatitis, colitis, hepatitis, and cirrhosis, where REG3A appears to function as part of a feedback mechanism aimed at dampening inflammation. Notably, similar patterns of dysregulation are observed in inflammation-associated cancers, such as pancreatic ductal adenocarcinoma, gastrointestinal malignancies, and primary liver cancer, raising the possibility that REG3A could serve as a diagnostic or prognostic biomarker. However, it remains difficult to determine whether REG3A actively promotes cancer or simply responds to changes in inflammation. This is further complicated by the fact that tumors can vary greatly over time and in different areas, and their genetic makeup can change as they grow. These factors underscore the need for large, well-characterized molecular cohorts to accurately assess REG3 protein expression patterns, a requirement that has not yet been fully met in current research.

Emerging evidence suggests that REG3 proteins may be co-opted by both malignant and stromal cells within the tumor microenvironment, where they could modulate antitumor immunity and oncogenic signaling. This potential functional reprogramming highlights the variable role of REG3A in cancer, influenced by factors such as tissue origin, cancer subtype and molecular profile, the pathological status of surrounding tissue, tumor stage, and the immune landscape.

Furthermore, some studies report a negative correlation between REG3A expression and tumor progression in specific settings [20,25,71,113], suggesting that REG3A may act as a tumor suppressor under certain conditions while promoting tumor growth in others. These discrepancies emphasize the highly variable and adaptable role of REG3A in cancer, shaped by the inflammatory milieu and surrounding signaling networks. Table 2 provides a comprehensive overview of the multifaceted role of REG3A in cancer, including a comparative analysis of key findings across various cancer types and experimental models, such as in vitro, in vivo, and human cohort studies.

Several limitations in the current literature hinder a clear understanding of its role in cancer. Most studies rely heavily on in vitro models using diverse cell lines, which may lack biological relevance for a secreted protein like REG3A that operates across multiple cell types and complex systems such as the gut microbiota and metabolism. This may contribute to experimental variability, poor reproducibility, and often contradictory findings. In vivo studies on REG3A remain limited, resulting in significant gaps in our understanding of its functional roles in mammalian cancer models. Notably, conclusions about its tumor-promoting or tumor-suppressive effects often rely on single studies, especially in colorectal, gastric, and liver cancers. To date, pancreatic cancer has received the most attention, with, to our knowledge, approximately ten in vivo studies conducted on three distinct cancer models (Table 2). A further challenge is distinguishing REG3A’s intracellular effects from its extracellular, autocrine, or paracrine functions. This mechanistic uncertainty complicates the interpretation of its biological activity and its role in tumor progression and microenvironmental interactions. Collectively, these limitations present a substantial obstacle to fully elucidating the relevance of REG3A in cancer and underscore the need for replication studies, physiologically relevant models, and integrative research approaches.

### 7.1. REG3A in Pancreatic Cancer: A Paracrine Oncogenic Role

REG3A is highly expressed in PDAC and is strongly associated with metastatic invasion, poor surgical outcomes, lower progression-free survival, and reduced overall life expectancy (Table 2) [48,77,114,115]. Its expression in pancreatic tissue is significantly elevated in PDAC (up to 24-fold compared to other pancreatic diseases, and 16-fold relative to chronic pancreatitis) [114]. Elevated serum REG3A levels are also detected in PDAC patients and serve as a diagnostic biomarker, distinguishing cases from healthy individuals with 90% sensitivity and 82.8% specificity at cutoff of 18 µg/L [115]. Validation in a cohort of 118 patients further confirmed the association of increased REG3A (both in serum and pancreatic juice) with poor prognosis [114].

Mechanistically, transcriptomic analyses of PDAC cell lines have shown that REG3A upregulates genes involved in cell proliferation and survival programs [116]. Notably, REG3A is primarily secreted by peritumoral pancreatic acinar cells rather than by tumor cells themselves [48,77,114,117]. Acting as a paracrine factor, it activates JAK/STAT and EXTL3/RAS/ERK signaling pathways in both inflamed acinar cells and malignant epithelial cells. In acinar cells, this promotes acinar-to-ductal metaplasia and the formation of pancreatic intraepithelial neoplasia (PanIN) lesions [76], while in cancer cells it enhances tumor growth and perineural invasion [61,70,98,99] (Figure 3). The oncogenic activity of REG3A is counterbalanced by SOCS3, a negative regulator of cytokine signaling. Silencing SOCS3 enhances REG3A-driven tumor progression, while its overexpression suppresses malignancy across multiple pancreatic cancer cell lines [112].

### 7.2. REG3A in Liver Cancer: A Tumor Suppressive Role Targeting the O-GlcNAc Glycosylation Pathway

REG3A was initially identified in hepatocellular carcinoma (HCC) in 1992 [3], but its role in liver carcinogenesis remained unclear until recently [20]. A number of studies have reported marked overexpression of REG3A in tumor tissue compared to adjacent non-tumor liver, validated at both mRNA and protein levels [27,74]. Elevated REG3A correlates with aggressive features such as larger tumor size and vascular invasion [118], and is found in 24–79% of HCC cases, depending on the detection method [3,27]. Elevated serum REG3A is also observed in 45% of cirrhotic patients and up to 75% with HCC [27]. REG3A expression is enriched in an HCC molecular subtype marked by Wnt/β-catenin activation and CTNNB1 mutations, often linked to early-stage tumors and better prognosis [60,73].

In vitro, REG3A enhances HCC cell proliferation, migration, and invasion via MAPK/ERK and PDGF-ββ signaling [74,118]. However, in vivo studies show a tumor-suppressive function. Moniaux et al. found that hepatic overexpression of REG3A significantly reduces tumor burden and extends survival in both chemical and genetic mouse models [20]. Mechanistically, REG3A reduces O-GlcNAcylation, including MYC modification, by reducing intracellular UDP-GlcNAc levels, thereby promoting MYC phosphorylation and degradation. This effect is independent of OGT, OGA, and GFAT expression or activity, and instead depends on the ability of REG3A to bind hexosamine pathway intermediates via its EPN lectin motif. A mutant (EPN→GPG) form fails to bind these metabolites or reduce O-GlcNAcylation, confirming the importance of carbohydrate-binding. Supporting this, mouse livers expressing wild-type REG3A and non-tumor tissues from HCC patients show reduced UDP-GlcNAc and global O-GlcNAcylation. Clinically, cirrhotic patients with high hepatic REG3A have longer cancer-free survival, although REG3A levels do not affect outcomes once HCC develops [20]. These results suggest that REG3A promotes tumor growth in simple in vitro systems but suppresses tumor formation in vivo by regulating O-GlcNAcylation, a process shaped by systemic metabolism and tissue interactions (Figure 3). These results emphasize the need for physiologically relevant in vivo models that accurately represent human tumors. REG3A exerts pleiotropic functions that extend beyond the tumor itself, influencing organ-level physiology, host–microbiota interactions, and metabolism, factors poorly represented in in vitro systems. Without models that incorporate native tissue architecture and immune components, interpretations of REG3A’s oncogenic potential remain incomplete, limiting progress in therapeutic translation.

### 7.3. The REG3A Paradox: Conflicting Data Across Solid Tumor Models

The expression of REG3A is dysregulated across multiple human solid tumors, yet its functional role in these cancers remains highly nuanced and varies depending on tumor type and biological conditions (Table 2). In cholangiocarcinoma, REG3A is markedly overexpressed in tumor tissues relative to adjacent non-neoplastic bile duct epithelium, suggesting a possible role in tumor initiation or maintenance [27]. Similarly, in bladder cancer, elevated urinary levels of REG3A have been reported in patients with advanced disease stages. These levels are not only associated with disease progression but also demonstrate promising diagnostic accuracy (sensitivity: 80.2%; specificity: 78.2%) [51]. In contrast, the prognostic and functional relevance of REG3A in lung and head and neck cancers appears more variable. In lung adenocarcinoma, REG3A expression shows no clear correlation with clinical parameters [119]. However, in hypopharyngeal squamous cell carcinoma, high REG3A expression is linked to prolonged patient survival and enhanced sensitivity to chemotherapeutic agents. Supporting these clinical observations, in vitro studies have demonstrated that REG3A exerts anti-proliferative effects and enhances cellular responses to both chemotherapy and radiotherapy [66,120]. The most compelling evidence for the dualistic nature of REG3A comes from studies in gastric and colorectal cancers, where its role oscillates between tumor-suppressive and oncogenic depending on experimental conditions and disease characteristics.

In gastric cancer (GC), several studies report a significant downregulation of REG3A in tumor tissues, implying a tumor-suppressive function [25,71,121]. Functional analyses further reveal that REG3A overexpression inhibits proliferation and induces apoptosis in gastric cancer cell lines, largely through modulation of the PI3K/Akt signaling pathway and upregulation of DMBT1, a known tumor suppressor [71,121]. Conversely, other studies describe elevated REG3A expression in GC tissues, where its knockdown results in decreased cancer cell proliferation, migration, and invasion, indicating a potential oncogenic role under certain conditions [72]. Of note, the sample sizes in the human cohort studies differed significantly: Chen et al. [72] analyzed 41 paired non-tumor/tumor gastric tissue samples, whereas Qiu et al. [71] conducted a large-scale analysis using TCGA, GSE13911, and GSE13861 datasets, encompassing a total of 876 patients with gastric cancer. To date, no further studies have been reported that could clarify the role of REG3A in gastric cancer.

Similar contradictions are observed in colorectal cancer (CRC). One large-scale analysis of 331 matched tumor–normal tissue pairs found REG3A to be enriched in tumors lacking deep invasion, with high expression correlating with improved overall survival [57]. These findings align with data from murine models of colitis-associated cancer, where the murine REG3A homolog suppresses cytokine-driven STAT3 activation, thereby limiting early neoplastic transformation and tumor progression [69]. However, other studies show that REG3A is overexpressed in advanced CRC and is linked to poorer clinical outcomes [70]. Both studies relied on large-scale cohorts, yet they reported opposing correlations between REG3A expression and cancer progression, one showing a negative correlation [57] and the other a positive association [70]. A key methodological difference lies in the choice of control groups: Zheng et al. used pairwise non-tumor tissue adjacent to the tumor [35], whereas Ye et al. compared tumor samples to colonic tissues from a limited number of healthy donors [70]. The imbalance in sample sizes between control and tumor groups (e.g., 6 healthy vs. 90 tumor tissues in GSE33113; 28 healthy vs. 245 tumor tissues in TCGA) may also contribute to the statistical differences found. Similarly, methodological differences in in vivo mouse models are notable. Liu et al. reported a tumor-suppressive role for REG3A, observing a reduced polyp burden in the AOM-DSS-induced colorectal cancer model [69]. In contrast, Ye et al. described a pro-tumoral effect of REG3A following ectopic transplantation of human cancer cells into the flanks of immunodeficient (nude) mice [70]. Mechanistically, using in vitro experiments on the same cancer cell lines, Ye et al. demonstrated that REG3A promotes cancer cell proliferation, migration, and invasion through activation of the AKT and ERK1/2 signaling pathways [70].

At the epigenetic and transcriptional levels, non-coding RNA regulation adds further layers of control. Notably, Yari et al. identified a long non-coding RNA (lncRNA REG1CP) transcribed from a pseudogene located between REG1A and REG1B on chromosome 2p12. REG1CP was shown to interact directly with the REG3A promoter via a triplex RNA/DNA structure, recruiting the helicase FANCJ and facilitating transcriptional activation under the control of the glucocorticoid receptor α [122]. REG1CP is notably upregulated in both preneoplastic colonic lesions and colorectal carcinomas, suggesting a cooperative axis between REG1CP and REG3A in promoting colorectal tumorigenesis.

Collectively, these findings highlight the nuanced and highly variable nature of REG3A’s involvement in cancer. Its function varies not only between tumor types but also within the same malignancy, depending on molecular subtype, tumor stage, and microenvironmental cues. While correlations between REG3A expression and clinical outcomes have been documented across multiple cancer types, mechanistic understanding remains incomplete. Elucidating the signaling networks and regulatory elements that govern REG3A expression and activity will be essential to determine its viability as a diagnostic biomarker, prognostic indicator, or therapeutic target. Given these complexities, future studies should prioritize the use of physiologically relevant in vivo models that preserve tissue-specific architecture, immune interactions, and systemic regulation. Only through such integrative approaches can the full spectrum of REG3A’s oncogenic and tumor-suppressive functions be defined, paving the way for informed translational applications in oncology.

### 7.4. The REG3A–Gut Microbiota–Inflammation–Cancer Axis: An Emerging Interface in Tumor Biology

Over the past two decades, mounting evidence has highlighted the critical role of the microbiota in cancer development, progression, and therapeutic response (reviewed in [123]). Once viewed as passive bystanders, microbial communities are now understood to be active modulators of host physiology, interacting with genetic, immune, metabolic, and environmental factors to influence carcinogenesis. Among these, the gut microbiome has been most extensively studied, particularly in colorectal cancer. Specific bacterial species, such as *Fusobacterium nucleatum*, colibactin-producing *Escherichia coli*, and enterotoxigenic *Bacteroides fragilis*, exhibit pro-tumorigenic properties [124,125]. These microbes can induce DNA damage, promote chronic inflammation, and reshape immune responses in ways that favor tumor initiation and growth. For example, *F. nucleatum* adheres to epithelial cells, activates oncogenic β-catenin signaling, and impairs antitumor immunity by recruiting myeloid-derived suppressor cells (MDSCs) and inhibiting natural killer (NK) cell activity [126].

Dysbiosis-driven chronic inflammation is a well-recognized driver of carcinogenesis. Microbial metabolites and ligands for pattern recognition receptors (e.g., Toll-like receptors) can amplify inflammatory responses and disrupt epithelial barriers. In inflammatory bowel disease, an established risk factor for CRC, microbial imbalances closely correlate with disease activity and barrier dysfunction, creating a tumor-permissive microenvironment.

Although direct evidence linking REG3A-modulated microbiota to cancer is currently lacking, REG3A’s known ability to shape a beneficial microbial community and preserve epithelial integrity suggests a potential protective role, particularly in colorectal and liver cancers [86]. REG3A promotes the maintenance of Firmicutes such as *Lachnospiraceae* and *Ruminococcaceae*, Gram-positive anaerobes that produce short-chain fatty acids (SCFAs) like butyrate [86]. These metabolites are generally considered protective, they reinforce epithelial barrier function, reduce inflammation, and may help suppress tumorigenesis [127].

Microbial influence on carcinogenesis extends beyond the gut. A well-established example is *Helicobacter pylori*, a major risk factor for gastric cancer, which contributes to tumorigenesis through chronic inflammation, epigenetic reprogramming, and suppression of tumor suppressor pathways, including those involving DMBT1 and E-cadherin. Notably, REG3A has been identified as a positive regulator of DMBT1, potentially counteracting H. pylori-mediated silencing of this tumor suppressor [128]. Whether this effect results from direct transcriptional regulation or is secondary to REG3A’s antimicrobial activity remains unresolved. The observed upregulation of REG3A in H. pylori-infected gastric tissue suggests a potential protective role, but further research is needed to clarify its function [129].

In summary, the REG3A-microbiota–cancer axis represents a promising but largely unexplored domain. While its impact on tumorigenesis remains to be definitively established, existing evidence suggests that REG3A may influence cancer risk by modulating microbial composition, inflammatory signaling, and epithelial integrity. Bridging this knowledge gap presents a valuable opportunity for future investigation.

## 8. Clinical Relevance and Therapeutic Perspectives

Despite a robust body of preclinical evidence supporting the involvement of REG3A in diverse pathological settings, including inflammation, tissue regeneration and metabolic disorders, its translation to clinical application remains nascent. To date, only a few clinical trials have investigated REG3A or related peptides in humans. Nevertheless, early-phase studies offer encouraging translational insights and suggest the potential for broader therapeutic utility. One such example involves INGAP peptide (INGAPpp), a structural homolog of REG3A derived from hamster, which has been evaluated in two randomized, placebo-controlled Phase II clinical trials targeting diabetes mellitus. In type 1 diabetes (T1D) (NCT00071409), 63 patients received daily subcutaneous injections of 300 mg or 600 mg INGAP for 90 days [106]. The primary outcome, arginine-stimulated C-peptide via mixed-meal tolerance test, showed a significant increase in the 600 mg group compared to placebo (*p* = 0.0058), indicating partial β-cell functional recovery. Secondary outcomes, including HbA1c and insulin dose, improved modestly. Approximately 50% of the treatment effect persisted after a 30-day washout, suggesting transient β-cell preservation. Injection-site reactions were common (up to 90%), but no serious systemic adverse events or immunogenic responses were reported. A parallel Phase II study in type 2 diabetes (T2D), involving 126 participants, confirmed these findings [106]. The 600 mg group demonstrated significantly elevated C-peptide levels (*p* = 0.031) and a 0.9% reduction in HbA1c at day 90 (vs. 0.47% in placebo, *p* = 0.009), with durable metabolic effects persisting at follow-up (*p* = 0.013). Side effects were again limited primarily to injection-site reactions (about 65%), without notable systemic toxicity.

A second approach has explored recombinant human REG3A protein (ALF 5755) in the treatment of acute liver injury. A Phase IIa randomized, double-blind, placebo-controlled trial (NCT01318525) enrolled 57 adults with severe acute hepatitis (SAH) or early-stage acute liver failure (ALF) of non-acetaminophen etiology [88]. ALF 5755 (5 mg/kg, IV) was evaluated for its impact on coagulation recovery (change in prothrombin ratio over 72 h). While overall results across all etiologies were neutral, a subgroup analysis in HBV and autoimmune hepatitis patients revealed a significantly improved prothrombin slope (*p* = 0.04) and a reduction in hospitalization duration (8 vs. 14 days, *p* = 0.02). Prior Phase I data confirmed that ALF 5755 was well-tolerated in healthy volunteers, with no serious adverse events, acceptable pharmacokinetics, and no immunogenicity [21].

Together, these studies provide first-in-human validation of the regenerative, anti-inflammatory and metabolic potential of REG3A-derived therapeutics. While clinical indications to date have focused on diabetes and ALF, REG3A’s broader biological properties, including epithelial repair, preservation of intestinal barrier integrity, and modulation of innate immunity and the gut microbiota, point to its therapeutic promise in other conditions, such as mucosal injury and chronic inflammatory diseases. Its potential relevance to oncology is particularly compelling, given REG3A’s roles in immune modulation, epithelial regeneration, and microenvironmental remodeling. However, due to its variable effects, such as evidence of pro-tumorigenic activity in certain malignancies, careful consideration of disease type, underlying mechanisms and target cell populations is essential. Rigorous preclinical and translational studies are needed to delineate its safety profile and therapeutic window before advancing REG3A-based interventions in cancer. Moreover, recent advances in cancer immunotherapy, alongside growing evidence that gut microbiota composition influences treatment efficacy, highlight a promising opportunity for REG3A-based microbiota modulation to enhance immunotherapeutic outcomes. By reinforcing epithelial integrity and restoring a favorable microbial milieu, REG3A could synergize with immune checkpoint inhibitors to strengthen antitumor immunity and improve patient response rates.

In summary, REG3A-based interventions are emerging as promising tools in regenerative medicine, metabolic disorders, and potentially oncology, supported by early clinical safety and efficacy signals. However, conflicting data and concerns about REG3A’s oncogenic potential have slowed progress in cancer applications. Future studies should focus on patient selection, dose optimization, long-term safety, and understanding tissue-specific effects to fully unlock the clinical benefits of REG3A across diseases.

## 9. Conclusions

REG3A is a multifaceted molecule with critical and diverse roles in both health and disease. Physiologically, it plays a central role in the gastrointestinal tract by supporting antimicrobial defense, promoting epithelial repair, and maintaining mucosal homeostasis. In contrast, its involvement in cancer is complex and highly variable, with REG3A frequently dysregulated and linked to chronic inflammation, immune evasion, and aberrant cell proliferation, factors that complicate efforts to define a universal role in tumorigenesis. Despite gaps in our understanding of its biology, the diverse functions of REG3A highlight its strong clinical relevance. Its immunomodulatory and regenerative properties position it as a promising target for treating inflammatory diseases characterized by epithelial damage and remodeling. In oncology, its dual role as either a tumor promoter or suppressor depends on tumor type, cellular features, and signals from the surrounding microenvironment. Mechanistically, REG3A influences key oncogenic signaling pathways and metabolic reprogramming, notably glucose metabolism essential for cancer cell survival and proliferation. Furthermore, by modulating host–microbiota interactions, REG3A contributes to shaping tumor immunity and systemic inflammation, broadening its relevance in cancer biology and targeted therapy.

Importantly, REG3A has completed Phase 1 and 2 clinical trials for acute liver failure and diabetes, demonstrating favorable safety profiles. Nonetheless, its development as a therapeutic agent requires rigorous safety evaluation, especially for systemic or long-term use. Unlocking the full clinical potential of REG3A will require integrative research approaches that combine mechanistic insight with appropriate translational models to guide safe and effective applications. This will pave the way for innovative therapies harnessing the unique biology of REG3A in both regenerative medicine and precision oncology.

## Figures and Tables

**Figure 1 cancers-17-02395-f001:**
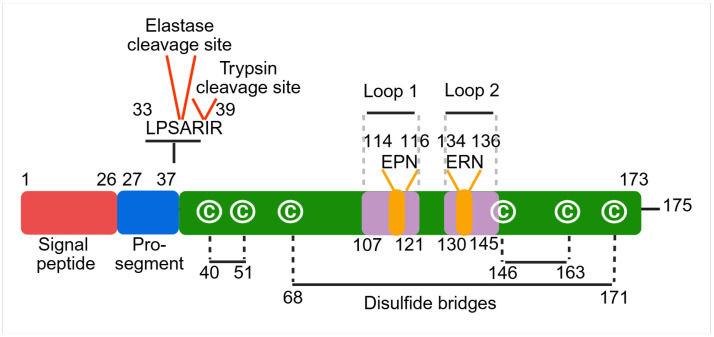
Domain structure of human REG3A. Schematic representation of the secondary structure of human REG3A (amino acids 1–175), highlighting the cleavage sites for trypsin and elastase, loops 1 and 2, and the three disulfide bridges. The green box denotes the mature REG3A lectin domain, which contains a single carbohydrate recognition domain (CRD). Cysteine residues involved in disulfide bonding are indicated by the letter C within circles. (Created in BioRender. https://BioRender.com/7r8q7cv (accessed on 12 June 2025).

**Figure 2 cancers-17-02395-f002:**
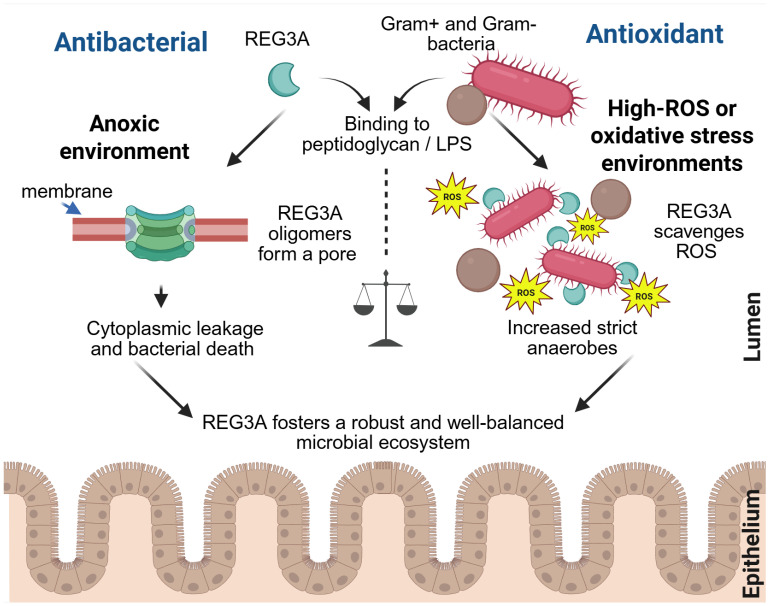
Dual antimicrobial and antioxidant functions of REG3A in the gut. Intestinal bacteria are shown in pink (bacilli) and brown (cocci). ROS: reactive oxygen species. (Created in BioRender. https://BioRender.com/nrtonjp (accessed on 16 June 2025).

**Figure 3 cancers-17-02395-f003:**
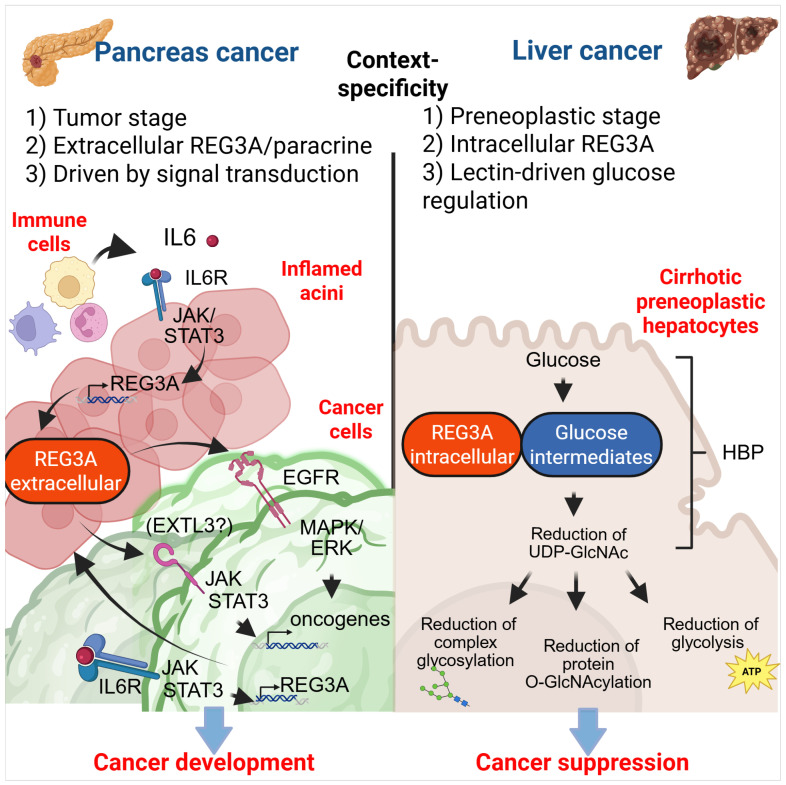
Context-dependent functions of REG3A in cancer. This schematic summarizes the dual roles of REG3A in cancer as an oncogene in pancreatic ductal adenocarcinoma (PDAC) and as a tumor suppressor in hepatocellular carcinoma (HCC). In PDAC, REG3A is induced by IL-6/JAK/STAT signaling in inflamed acinar cells (pink) adjacent to tumor lesions (green), promoting tumor progression via EGFR (and possibly EXTL3) activation and downstream pathways. In HCC, REG3A is expressed in early-stage hepatocytes, where it interacts with hexosamine biosynthesis pathway intermediates, lowering UDP-GlcNAc levels and altering metabolism to delay tumor development. (Created in BioRender. https://BioRender.com/lf76afg (accessed on 12 June 2025).

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
