# Peer review of "REG3A: A Multifunctional Antioxidant Lectin at the Crossroads of Microbiota Regulation, Inflammation, and Cancer"

_cancers, 2025, doi:10.3390/cancers17142395_

Round 1

Reviewer 1 Report

Comments and Suggestions for Authors

1] Can the authors provide more specific information about the proteolytic activation pathway, including the kinetics of trypsin and elastase cleavage and the relative biological activities of different cleavage products?

2] What is the molecular basis for REG3A's selective bactericidal activity against Gram-positive bacteria versus its limited effects on Gram-negative organisms?

3] Consider adding a comprehensive figure showing REG3A's domain structure, highlighting the EPN and QPN motifs, disulfide bridges, and activation sites for better visual understanding.

4] Develop a schematic diagram illustrating the dual antimicrobial and antioxidant functions of REG3A in the intestinal environment.

5] Clarify the difference between "antimicrobial peptide" and "lectin" functions of REG3A, as the paper sometimes uses these terms interchangeably.

6] What are the safety considerations and potential side effects of REG3A-based therapies, particularly given its role in both tumour suppression and tumour promotion?

7] What is the impact of diet on REG3A expression and subsequent microbiome composition?

8] Create a comparative table showing REG3A expression levels across different tissues and disease states.

9] Line 82-84, what might influence the second cleavage, and how does it relate to its biological activity?

Author Response

We sincerely thank the Reviewers and Editors for their positive feedback and thoughtful evaluation of our review. As outlined in our point-by-point response, we have carefully addressed all comments and revised the manuscript accordingly.

Reviewer#1. Comments and Suggestions for Authors

Comment 1. Can the authors provide more specific information about the proteolytic activation pathway, including the kinetics of trypsin and elastase cleavage and the relative biological activities of different cleavage products?

Response. Done. We have revised the text to provide a more detailed account of the current understanding of the proteolytic activation pathway.

Comment 2.  What is the molecular basis for REG3A's selective bactericidal activity against Gram-positive bacteria versus its limited effects on Gram-negative organisms?

Response. Thank you for this question. We have revised the text to address this complex and still largely unresolved question.

Comment 3. Consider adding a comprehensive figure showing REG3A's domain structure, highlighting the EPN and QPN motifs, disulfide bridges, and activation sites for better visual understanding.

Response. Done. We have added a new figure (Fig. 1) illustrating the domain structure of REG3A to support our description.

Comment 4. Develop a schematic diagram illustrating the dual antimicrobial and antioxidant functions of REG3A in the intestinal environment.

Response. We have added a new figure (Fig. 2) that illustrates the dual antimicrobial and antioxidant functions of REG3A in the gut. Thank you for this helpful suggestion.

Comment 5. Clarify the difference between "antimicrobial peptide" and "lectin" functions of REG3A, as the paper sometimes uses these terms interchangeably.

Response. REG3A is not, strictly speaking, an antimicrobial peptide, which is generally defined as a molecule containing fewer than 50 amino acids. REG3A is instead a C-type lectin that exhibits antimicrobial activity and is more accurately described as an antimicrobial protein. We have revised the text accordingly and removed references to REG3A as an antimicrobial peptide.

Comment 6. What are the safety considerations and potential side effects of REG3A-based therapies, particularly given its role in both tumour suppression and tumour promotion?

Response. Thank you for raising this important and complex question. We have included a discussion of the potential safety considerations and dual roles of REG3A in cancer in the final paragraph of the “Conclusions” section.

Comment 7. What is the impact of diet on REG3A expression and subsequent microbiome composition?

Response. We have added a new paragraph, along with relevant references, discussing the diet-sensitive regulation of REG3A expression and its impact on gut microbial ecology. Thank you for this comment.

Comment 8.  Create a comparative table showing REG3A expression levels across different tissues and disease states.

Response. Done.

Comment 9.  Line 82-84, what might influence the second cleavage, and how does it relate to its biological activity?

Response. As noted in our response to Comment 1, the second cleavage is influenced by several factors (host or bacterial proteases) and plays a key role in modulating the biological activity of REG3A.

Reviewer 2 Report

Comments and Suggestions for Authors

The paper "Reg3A: A Multifunctional Antioxidant Lectin at the Crossroads of Microbiota Regulation, Inflammation, and Cancer" highlights how REG3A helps keep tissues healthy and affects cancer growth, providing a detailed and scholarly overview of its many functions. A strong contender for publication in Cancers, pending minor but significant revisions, the work is rich in molecular information, widely cited, and carefully organized.

The scope of the work is obvious and relatively reasonable. The work examines REG3A's molecular, immunological, metabolic, and carcinogenic roles from its evolutionary beginnings. The writers present a logical and coherent narrative that begins with the construction of REG3A and concludes with its dual functions in both tumor suppression and promotion. The review moves from generic physiology to context-specific cancer processes, therefore balancing breadth and depth. The sections that explain how REG3A helps manage oxidative stress, its signaling through JAK/STAT and MAPK pathways, and its role in liver cancer through O-GlcNAcylation are particularly well-supported by strong experimental data from both lab and live models.

With almost 110 references from peer-reviewed studies and original material, the referencing is thorough and exact. Crucially, the references are from both experimental and clinical sources and are varied and current (many from 2023–2025). There is no clear overreliance on self-citation; even the group's past research is well-placed within the general literature. Citations adequately support all key statements; if results contradict one another (as in the case of colorectal and stomach tumors), the writers present the data fairly on both sides. This comprehensive technique highlights the complexity of REG3A's operation and lends credence to the results.

Methodologically, the choice and synthesis of the studies are strong, even if the publication is a narrative review rather than a systematic one. The authors show that they understand the limitations in the current research, especially the heavy dependence on in vitro data and the absence of relevant animal models in pancreatic and liver cancer studies (pages 10–11). This introspection enhances the scholarly value of the work and lends credibility to the suggested lines of action for the following study. The writers rightly point out that in vitro results cannot adequately reflect the several functions played by REG3A in dynamic microenvironments and immunological settings.

The results and analysis fit together quite nicely. The writers eloquently state the differences between the intracellular and extracellular actions of REG3A, particularly in liver and pancreatic models. The graphical abstract on page 9 (Figure 1) does a great job of showing the different roles of REG3A in pancreatic cancer (where it promotes cancer) and liver cancer (where it helps fight cancer). For accessibility, the figure would benefit from more color contrast and clearer legends.

Appropriately cautious, the last page notes both the clinical promise of REG3A and the present uncertainty surrounding its exact function in cancer. It presents a future research plan without exaggerating the translational readiness of present discoveries. To increase their impact, the conclusions might, however, be somewhat more concise and better organized into key lessons.

Although the paper would benefit from some editing for clarity and concision, the language throughout the work is professional and readable. We should simplify many sections to avoid repetition—for example, we cover the antioxidant properties of REG3A in detail in both sections 1 and 4. The paper repeatedly revisits the JAK/STAT pathway, sometimes using redundant terminology. We should consolidate these repetitions for improved flow, as they can potentially change the meanings and eliminate them.
All in all, this study is a relevant, well-researched, and well-argued assessment of the various biological functions of REG3A. It effectively combines several lines of data to create a coherent framework for grappling with the complexity of REG3A in health and disease. The paper is ready for publication with minimal changes to eliminate duplicates, enhance figure display, and refine wording. The editorial advice recommends accepting the paper with minor modifications.

Author Response

We appreciate your positive feedback and are glad that you found the review valuable.

 Comment 1. For accessibility, the figure would benefit from more color contrast and clearer legends.

Response. We have adjusted the color contrast and revised the legends to improve the figure’s accessibility and clarity.

Comment 2. the conclusions might, however, be somewhat more concise and better organized into key lessons.

Response. Thank you for this valuable suggestion. We have revised the conclusion to enhance clarity and organization, presenting the key messages in a more concise and structured manner.

Comment 3. We should simplify many sections to avoid repetition—for example, we cover the antioxidant properties of REG3A in detail in both sections 1 and 4. The paper repeatedly revisits the JAK/STAT pathway, sometimes using redundant terminology. We should consolidate these repetitions for improved flow, as they can potentially change the meanings and eliminate them.

Response: Thank you for pointing this out. We apologize for the redundancies related to REG3A’s antioxidant properties and the repeated references to the JAK/STAT pathway. We have revised the manuscript to eliminate these repetitions and improve overall flow and clarity.

Round 2

Reviewer 1 Report

Comments and Suggestions for Authors

Since the Authors addressed all the comments, the manuscript may be accepted 

Author Response

Thank you again for your favorable evaluation of our manuscript.

Reviewer 2 Report

Comments and Suggestions for Authors

Areas of Enhancement: Particularly regarding its antioxidant, antibacterial, and immunomodulating activities, the paper features parts that repeat the exact functional traits of REG3A. Often with overlapping references, these topics reappear in several sections—e.g., inflammation, regeneration, bacteria, and cancer. By grouping these tasks under a single biological mechanism section, a better structure would minimize repetition and maintain detail. For example, consolidation would help to distribute and profit from discussions in Sections 3, 4, and 5 about the ROS-scavenging ability of REG3A and its cytoprotective properties. Although the paper presents a wealth of tissue-specific and condition-specific data—e.g., Table 1 on p. 5 and subsequent cancer-specific discussions—often without fully integrating this knowledge into broader biological or therapeutic contexts. For instance, the consequences of REG3A's multiple roles in cancer are often discussed but not combined into a coherent mechanistic paradigm.

I can be improved by a visual model or decision tree compiling the context-specific pro- or anti-tumor effects of REG3A. Although the review addresses REG3A's interactions with immune cells, epithelial tissues, and bacteria, the intracellular signaling processes are sometimes presented as scattered ideas rather than a coherent framework. For example, although the EXTL3 and gp130/STAT3 pathways are discussed (pp. 8–9), a comparison of these pathways across multiple tissues or disease stages is lacking.

Interesting and deserving of further discussion is the role of REG3A in modulating metabolic reprogramming (e.g., UDP-GlcNAc regulation), which could be explored in a dedicated paragraph and accompanied by a schematic diagram.

Not all references, nevertheless, are favorably evaluated in terms of experimental design or model applicability. For instance, some conclusions depend mainly on in vitro studies but do not precisely distinguish them from in vivo or clinical findings (such as the various functions of REG3A in tumor suppression and cancer progression in gastric or colorectal cancer). Emphasizing when results, based on in vivo data or clinical studies, match those of preclinical models would assist readers in understanding the validity of each assertion.

Demand consistency in language use and clarity in terminology. The paper rightly discusses the uncertainty between REG3A and its murine orthologs (Reg3β, Reg3γ), although these differences are sometimes confused throughout. Studies obtained from mouse models must indicate whether functional extrapolation to human REG3A is confirmed or hypothetical. For translational significance, particularly in the sections on cancer and metabolic illnesses, this is vital.

Visual Figures and Diagrams— Ideas for Development: Although Figure 2 on page 6 is aesthetically pleasing, it could be improved by incorporating a schematic of either epithelium healing or downstream signaling pathways in immune cells. Figure 3 on page 11, which illustrates the context-dependent activities of REG3A in cancer, is effective but could benefit from a legend that highlights specific ligands or receptors involved and distinguishes between intracellular and extracellular actions.

The review did not thoroughly investigate the potential of REG3A as a druggable target or its pharmacological modulation using small molecules, biologics, or microbiome-derived therapies.

Furthermore, given the continuous studies (such as ALF-5755 for acute liver failure), focusing more on current research projects or patents will significantly increase the practical relevance of the results. Although a few parts may be tightened for clarity, the language is generally fluent and scientifically adequate. To get a more objective tone, we can change lines like "it appears to us" (p. 12).

Readability would be much improved by consistency in tense and vocabulary (e.g., "tumor-suppressive role" vs. "tumor suppressor function"). In general, this review is a well-crafted intellectual work that appears suitable for publication after minor revisions. As a vital immunometabolic and oncogenic modulator, REG3A makes a compelling argument.

Reduced redundancy, simpler mechanistic models, improved integration of conflicting results, and a greater focus on translational relevance would all help, though. These changes will make the paper a basic reference for the subsequent studies on REG3A and therapeutic development.

Author Response

Dear Reviewer #2,

We thank you for your time and thoughtful review of our manuscript, "Reg3A: A Multifunctional Antioxidant Lectin at the Crossroads of Microbiota Regulation, Inflammation, and Cancer." We appreciate your detailed comments and suggestions. However, we would like to respectfully express our surprise at the marked difference between your initial review and the current set of recommendations.

In your earlier assessment, you described the manuscript as a well-organized, data-rich, and publication-ready article requiring only minor editorial improvements and clarifications. You emphasized its scientific value, narrative coherence, robust referencing, and methodological clarity, concluding that the article was ready for publication with minimal modifications.

In contrast, your second review takes a considerably more critical position, recommending substantial structural and conceptual changes. Some points now raised, including concerns about clarity, terminology, and mechanistic integration, appear to contradict prior positive evaluations of those same elements. While we fully welcome constructive critique, the significant shift in feedback, without corresponding changes in the manuscript’s core content, has introduced uncertainty regarding expectations for final acceptance.

To ensure full transparency, we summarize below the key revisions we have already implemented in direct response to the previous round of feedback:

  1. We streamlined and consolidated the sections on antioxidant activity and signaling pathways to improve clarity and reduce redundancy.
  2. We revised figure captions, improved visual contrast, and added new figures and a table, in line with your and Reviewer #1’s suggestions.
  3. We restructured the conclusion to enhance clarity and emphasize key take-home messages.

We remain committed to further improving the manuscript but would like to respectfully raise a few points of disagreement regarding the new recommendations:

(i) The suggestion to reorganize the manuscript into new mechanistic sections and introduce schematic decision trees would entail a full conceptual rewrite. This contrasts sharply with your initial comment that only minor edits were required and would impose a substantial delay without clear editorial guidance mandating such a transformation.

(ii) The current version defines and consistently distinguishes REG3A from its murine orthologs (Reg3β, Reg3γ). The distinction between “tumor suppressor role” and “tumor suppressor function” seems stylistic rather than substantive. These concerns were not raised in your initial review.

(iii) The referencing strategy, previously commended for rigor and balance, has now been downgraded in rating (from four to two stars) without specific justification. This also applies to ratings across other evaluated categories, despite limited content change.

(iv) We agree that clearly distinguishing in vitro and in vivo findings is important, and we believe the manuscript already makes these distinctions where appropriate. This issue was not raised in the initial round of comments.

(v) You now recommend expanding the discussion on REG3A as a therapeutic target, including mention of programs or patents such as ALF-5755. However, as stated in the manuscript, these developments are not focused on cancer, but rather on acute liver failure and diabetes. While we did incorporate some safety-related insights after the first review, we believe further elaboration would extend beyond the scope of a mechanistically oriented biology review.

We respectfully request clarification regarding the editorial priority of these new recommendations, specifically, which points are considered essential for acceptance versus those that are discretionary or speculative.

For example, suggestions such as unifying the diverse roles of REG3A in cancer into a "coherent mechanistic paradigm," or encouraging discussion of hypothetical pharmacologic modulation strategies (e.g., via small molecules or microbiome-derived therapies) not yet supported by existing preclinical or clinical data, seem to go beyond the bounds of available evidence and the intended scope of our review.

We remain entirely open to further revisions and collaborative refinement of the manuscript, but given the significant shift in your recommendations, we would greatly appreciate guidance to align our efforts with the journal’s editorial expectations.

Thank you again for your time, expertise, and engagement with our work. We value your contributions to the review process and welcome any further clarification you are willing to provide.

Sincerely,
Prof. Jamila Faivre (on behalf of all co-authors)

Round 3

Reviewer 2 Report

Comments and Suggestions for Authors

The manuscript "REG3A: A Multifunctional Antioxidant Lectin at the Crossroads of Microbiota Regulation, Inflammation, and Cancer" is an ambitious scientific review that examines the diverse biological roles of REG3A in immunity, oxidative stress, metabolic regulation, and cancer growth. The research demonstrates considerable depth and breadth; however, there are a few essential aspects that require improvement before it can be published in Cancers or another high-impact oncology journal.

1. Mechanistic explanations that aren't always the same depth: The manuscript discusses various functions and chemical pathways associated with REG3A; however, in certain sections, it lacks detailed explanations of how these processes work. For instance, REG3A's role as an antioxidant in acute liver failure is supported by its ability to scavenge ROS and bind to fibrin. However, the manuscript lacks a detailed explanation of the downstream molecular signaling, such as altering Nrf2, GSH, or other antioxidant networks. In the cancer sections, especially for colorectal and gastric cancers, the manuscript talks about results that don't match up but doesn't explain much about the signaling networks, the experimental setups, or the genetic changes that might clarify these differences.

Suggestion: Add a more thorough discussion of the conflicting evidence, using examples from how experiments were set up (like using 2D versus 3D cultures and missing parts of the tumor environment). This would make the paper more transparent and more scientific.

2. Too much reliance on in vitro studies: The publication repeatedly uses in vitro results to support its claims about REG3A's roles in tissue healing and cancer growth. However, there aren't enough in vivo studies, especially those that utilize physiologically relevant animal models or patient-derived xenografts. Mentioning them only serves to criticize the results, not to provide support.

Suggestion: Create a structured table that compares in vitro and in vivo results from different cancer models (pancreatic, liver, colorectal) and highlights where gaps exist and where the results are credible. The review should evaluate the results based on the models' applicability in real-life scenarios.

3. The dual role of REG3A in cancer is unclear: One of the main ideas is that REG3A can both benefit and harm tumors. Despite acknowledging this complexity, the current manuscript lacks a unifying framework or suggested model to describe it.

Suggestion: Make a clear diagram or model that shows how REG3A levels vary, where it is located in the cell, what type of cell it comes from (like tumor or stromal), and the related immune or metabolic situation. The picture on page 11 initiates this process, but it lacks sufficient detail and notes to explain it for all types of cancer.

4. Repeated information in the expression profile description: The long list of REG3A expression in different tissues and disorders is helpful; however, it is repeated word for word in both the narrative and Table 1 (p. 5). For example, the fact that REG3A is overexpressed in HCC, PDAC, and colorectal cancer is repeated in three sections using similar phrasing.

Recommendation: Make the story easier to read by using Table 1 when it makes sense and getting rid of language that is the same. Focus on patterns of expression that are truly diagnostic or prognostic, and discuss them in separate sections.

5. Poor integration of the microbiota-immune-tumor axis: One of the more original contributions this paper makes is the connection between REG3A's effect on microbiota and its impact on the whole body's metabolism and cancer. But the manuscript doesn't always stick to this idea.

Recommendation: Create a section called "REG3A–Microbiota–Inflammation–Cancer Axis," using new research that shows how substances from microbes, the health of barriers in the body, and immune system balance help tumors grow. This would support the overall framework suggested in the title.

6. Citation practices that aren't fair: The manuscript has many references, but some statements, like "REG3A acts as a metabolic modulator in multiple organs" or "REG3A modulates immune surveillance," are backed by just one source or no experimental evidence at all. On the other hand, multiple citations support specific claims, confirming the same point.

Suggestion: Recheck the citation list to make sure it's balanced, especially for strong statements. Additionally, include new studies from 2022 to 2025 that have a significant impact, demonstrating that science is still advancing.

7. The Clinical Relevance Section Needs More Work: The report briefly discusses Phase 1 and 2 studies for REG3A (p. 14) but doesn't provide detailed information on outcomes, patient groups, or clinical problems.

Suggestion: Create a lengthy paragraph that discusses the current state of clinical development, including trial identifiers, therapeutic indications (such as liver failure or diabetes), and the potential for repositioning REG3A in oncology or regenerative medicine.

Final Suggestion: Big Change

The work provides a thorough and promising summary of REG3A; however, it falls short in terms of critical analysis, mechanistic coherence, and translational framing in its current form. We need a major redesign that prioritizes the following areas:

• Making mechanistic ambiguity more straightforward

• Making translational insight stronger

• Making graphical and tabular synthesis better

• Getting rid of redundancies and unbalanced references

With these changes, the manuscript could become the definitive source of information on the complex biology of REG3A in the fields of immunology, metabolism, and cancer.

Author Response

Comments and Suggestions for Authors

Reviewer #2

The manuscript "REG3A: A Multifunctional Antioxidant Lectin at the Crossroads of Microbiota Regulation, Inflammation, and Cancer" is an ambitious scientific review that examines the diverse biological roles of REG3A in immunity, oxidative stress, metabolic regulation, and cancer growth. The research demonstrates considerable depth and breadth; however, there are a few essential aspects that require improvement before it can be published in Cancers or another high-impact oncology journal.

We sincerely thank Reviewer #2 for the thoughtful and constructive feedback, which has significantly contributed to improving our manuscript. We have carefully addressed all comments and hope that our revisions meet the expectations of the Reviewers and Editors.

Comment 1. Mechanistic explanations that aren't always the same depth: The manuscript discusses various functions and chemical pathways associated with REG3A; however, in certain sections, it lacks detailed explanations of how these processes work. For instance, REG3A's role as an antioxidant in acute liver failure is supported by its ability to scavenge ROS and bind to fibrin. However, the manuscript lacks a detailed explanation of the downstream molecular signaling, such as altering Nrf2, GSH, or other antioxidant networks.

Response 1. We have revised the text in the Introduction section as follows: Beyond its lectin functions, REG3A has emerged as a potent reactive oxygen species (ROS) scavenger, with a strong affinity for hydroxyl radicals [21]. Accumulating evidence supports its role in protecting tissues against oxidative injury. In acute liver failure, REG3A binds to fibrin scaffolds in necrotic areas of the liver, where it reduces ROS levels and promotes hepatocyte survival [22]. In muscle cells, REG3A prevents oxidative damage to the glycoprotein GP130, thereby preserving GP130/AMPK signaling and enhancing glucose uptake [23]. However, the underlying antioxidant mechanisms remain incompletely understood, particularly whether REG3A engages with canonical redox pathways such as Nrf2 signaling. Given the pivotal role of Nrf2 in regulating antioxidant responses, it is plausible that REG3A may influence or interact with this pathway. Further studies are warranted to determine whether REG3A modulates Nrf2 activity, either directly or indirectly through its effect on oxidative stress, as well as its potential impact on other antioxidant mediators, including superoxide dismutase (SOD), catalase and heme oxygenase-1 (HO-1).”

Comment 2. In the cancer sections, especially for colorectal and gastric cancers, the manuscript talks about results that don't match up but doesn't explain much about the signaling networks, the experimental setups, or the genetic changes that might clarify these differences. Suggestion: Add a more thorough discussion of the conflicting evidence, using examples from how experiments were set up (like using 2D versus 3D cultures and missing parts of the tumor environment). This would make the paper more transparent and more scientific.

Response 2. In response to Reviewer#2, we have expanded our discussion in the section 7.3. “The REG3A paradox: conflicting data across solid tumor models” to include more detailed explanations of methodological differences (e.g., cohort sizes, control groups, in vivo models) that may account for the conflicting results, as shown in the following additions:
Regarding gastric cancer, the text was amended by the following paragraph: “Of note, the sample sizes in the human cohort studies differed significantly: Chen et al. [105] analyzed 41 paired non-tumor/tumor gastric tissue samples, whereas Qiu et al. [107] conducted a large-scale analysis using TCGA, GSE13911 and GSE13861 datasets, encompassing a total of 876 patients. To date, no further studies have been reported that could clarify the role of REG3A in gastric cancer.” In colorectal cancer, the text was amended by the following paragraph: “Both studies relied on large-scale cohorts, yet they reported opposing correlations between REG3A expression and cancer progression, one showing a negative correlation [57] and the other a positive association [110]. A key methodological difference lies in the choice of control groups: Zheng et al. used pairwise non-tumor tissue adjacent to the tumor, whereas Ye et al. compared tumor samples to colonic tissues from a limited number of healthy donors. The imbalance in sample size between control and tumor groups (e.g., 6 healthy vs. 90 tumor tissues in GSE33113; 28 healthy vs. 245 tumor tissues in TCGA) may also contribute to the statistical differences found. Similarly, methodological differences in in vivo mouse models are notable. Liu et al. reported a tumor-suppressive role for REG3A, observing a reduced polyp burden in the AOM-DSS-induced colorectal cancer model [109]. In contrast, Ye et al. described a pro-tumoral effect of REG3A following ectopic transplantation of human cancer cells into the flanks of immunodeficient (nude) mice [110]. Mechanistically, using in vitro experiments on the same cancer cell lines, Ye et al. demonstrated that REG3A promotes cancer cell proliferation, migration, and invasion through activation of the AKT and ERK1/2 signaling pathways. [106]. “

Comment 3. Too much reliance on in vitro studies: The publication repeatedly uses in vitro results to support its claims about REG3A's roles in tissue healing and cancer growth. However, there aren't enough in vivo studies, especially those that utilize physiologically relevant animal models or patient-derived xenografts. Mentioning them only serves to criticize the results, not to provide support. Suggestion: Create a structured table that compares in vitro and in vivo results from different cancer models (pancreatic, liver, colorectal) and highlights where gaps exist and where the results are credible. The review should evaluate the results based on the models' applicability in real-life scenarios.

Response 3. To offer a comprehensive overview of the multifaceted role of REG3A in cancer, this review includes a new table (Table 2) that presents a comparative analysis of key findings across various cancer types and experimental models, including in vitro, in vivo, and human cohort studies. These primary findings are further synthesized and discussed in narrative form in Sections 7.1 to 7.3. In view of Table 2, we have modified our statement as follows: « Table 2 provides a comprehensive overview of the multifaceted role of REG3A in cancer, incorporating a comparative analysis of key findings across various cancer types and experimental models, including in vitro, in vivo and human cohort studies. Several limitations in the current literature hinder a clear understanding of its role in cancer. Most studies rely heavily on in vitro models using diverse cell lines, which may lack biological relevance for a secreted protein like REG3A that operates across multiple cell types and complex systems such as the gut microbiota and metabolism. This may contribute to experimental variability, poor reproducibility, and often contradictory findings. In vivo studies on REG3A remain limited, resulting in significant gaps in our understanding of its functional roles in mammalian cancer models. Notably, conclusions about its tumor-promoting or tumor-suppressive effects often rely on single studies, especially in colorectal, gastric, and liver cancers. To date, pancreatic cancer has received the most attention, with, to our knowledge, approximately ten in vivo studies conducted on three distinct cancer models (Table 2). A further challenge is distinguishing REG3A’s intracellular effects from its extracellular, autocrine or paracrine functions. This mechanistic uncertainty complicates the interpretation of its biological activity and its role in tumor progression and microenvironmental interactions. Collectively, these limitations present a substantial obstacle to fully elucidating the relevance of REG3A in cancer and underscore the need for repli-cation studies, physiologically relevant models, and integrative research approaches.

Thanks to Reviewer #2 for their suggestion and for encouraging us to look more closely at the gaps that need to be addressed.

Comment 4. The dual role of REG3A in cancer is unclear: One of the main ideas is that REG3A can both benefit and harm tumors. Despite acknowledging this complexity, the current manuscript lacks a unifying framework or suggested model to describe it. Suggestion: Make a clear diagram or model that shows how REG3A levels vary, where it is located in the cell, what type of cell it comes from (like tumor or stromal), and the related immune or metabolic situation. The picture on page 11 initiates this process, but it lacks sufficient detail and notes to explain it for all types of cancer.

Response 4. We appreciate Reviewer’s #2 thoughtful comment and agree that the dual role of REG3A in cancer remains a complex and unresolved issue. A major challenge in establishing a unifying model lies in the limited availability of precise data, particularly regarding cellular and subcellular localization of REG3A in relation to its context-dependent biological activity. In the manuscript, we outline several key factors that may contribute to this complexity and warrant further investigation. These include: (i) The dual involvement of REG3A in inflammation and tumorigenesis, which complicates its interpretation in cancers arising from chronic inflammatory conditions, such as hepatocellular carcinoma and pancreatic ductal adenocarcinoma. (ii) Its subcellular localization (e.g., intracellular vs. secreted forms), which may influence its mode of action. (iii) The specific inflammatory and signaling pathways engaged by REG3A, which vary by context. (iv) Tumor heterogeneity and the evolving tumor microenvironment, both of which may alter REG3A expression and distribution over time. Unfortunately, given the current limitations of the available literature, we are unable to provide a more comprehensive model than what is depicted in Figure 3. This figure highlights the contrasting roles of REG3A in the liver and pancreas, two organs in which its functions have been most extensively studied. In pancreatic ductal adenocarcinoma (PDAC), REG3A is secreted by peritumoral acinar cells and acts in a paracrine manner on tumor cells through the EGFR/JAK/STAT3 or EXTL3/RAS/ERK signaling pathways, thereby promoting tumor progression. In contrast, hepatocellular carcinoma (HCC) typically arises in the setting of chronic inflammation, such as cirrhosis, where hepatocytes begin expressing REG3A before tumor development. In this preneoplastic context, REG3A acts as a cytosolic lectin, that binds glucose derivatives from the hexosamine biosynthesis pathway, leading to reduced UDP-GlcNAc production and global O-GlcNAcylation. Since O-GlcNAcylation promotes liver carcinogenesis by activating oncogenes such as c-MYC and KRAS, the inhibition of this pathway by REG3A supports its tumor-suppressive role in HCC. The revised schematic on page 11 illustrates these context-dependent roles based on the most well-characterized examples.

Comment 5. Repeated information in the expression profile description: The long list of REG3A expression in different tissues and disorders is helpful; however, it is repeated word for word in both the narrative and Table 1 (p. 5). For example, the fact that REG3A is overexpressed in HCC, PDAC, and colorectal cancer is repeated in three sections using similar phrasing. Recommendation: Make the story easier to read by using Table 1 when it makes sense and getting rid of language that is the same. Focus on patterns of expression that are truly diagnostic or prognostic, and discuss them in separate sections.

Response 5. We have deleted the paragraph summarizing the expression of REG3A and summarized the expression profile in a table (Table 1). This recommendation was made by Reviewers #1 and #2, whom we thank for their suggestion. The text has also been modified by adding a new paragraph presenting the diagnostic and prognostic value of REG3A in several pathological conditions. The text can now be read as follows:

 “REG3A expression is markedly upregulated in response to both acute and chronic tissue stress, as well as across a range of malignancies. A comprehensive summary of these expression patterns is provided in Tables 1 and 2. Owing to its strong diagnostic relevance, REG3A has gained recognition as a circulating biomarker in several pathological conditions, where elevated serum levels often correlate with disease severity and poor prognosis. In chronic colitis [45] , gastrointestinal graft-versus-host disease (GI-GvHD) [46], sepsis [47], and pancreatic ductal adenocarcinoma (PDAC) [48], increased REG3A levels reflect heightened disease activity. Notably, in GI-GvHD, plasma REG3A concentrations are significantly higher in affected patients compared to those without gastrointestinal involvement. A threshold of ≥ 72 ng/mL has been shown to predict a 1.9-fold increased risk of non-relapse mortality, independent of other clinical variables, establishing REG3A as a validated prognostic marker in this setting [46,49]. In sepsis, REG3A has recently emerged as both a diagnostic and prognostic biomarker. Serum levels correlate strongly with clinical severity scores (SOFA and APACHE II), accurately distinguish septic patients from healthy controls, and independently predict 28-day survival [47]. Similarly, in rectal cancer patients undergoing neoadjuvant chemoradiotherapy (CCRT), transcriptomic analyses identified REG3A as the most upregulated gene in non-responders (log₂ ≈ 1.25, p = 0.0079). High REG3A protein expression is associated with poor treatment response, increased lymph node metastasis, vascular invasion, and serves as an independent predictor of both disease-specific and metastasis-free survival [50]. Furthermore, REG3A shows promising utility as a urinary biomarker in bladder cancer. In a single-cohort study, REG3A demonstrated a sensitivity of 80.2%, specificity of 78.2%, positive predictive value of 75.7%, negative predictive value of 82.3%, and an area under the curve (AUC) of 0.863, outperforming conventional markers such as NMP22 (sensitivity 52.1%, specificity 93.5%) [51]. Collectively, these findings underscore emerging clinical value of REG3A as a biomarker in both inflammatory and oncologic gastrointestinal diseases.”

Comment 6. Poor integration of the microbiota-immune-tumor axis: One of the more original contributions this paper makes is the connection between REG3A's effect on microbiota and its impact on the whole body's metabolism and cancer. But the manuscript doesn't always stick to this idea. Recommendation: Create a section called "REG3A–Microbiota–Inflammation–Cancer Axis," using new research that shows how substances from microbes, the health of barriers in the body, and immune system balance help tumors grow. This would support the overall framework suggested in the title.

Response 6. That's an excellent idea. Thank you very much. Below is the text of the new paragraph dealing with this point.

7.4. The REG3A-Gut Microbiota-Inflammation-Cancer Axis: an emerging interface in tumor biology

Over the past two decades, mounting evidence has highlighted the critical role of the microbiota in cancer development, progression and therapeutic response (reviewed in [125]). Once viewed as passive bystanders, microbial communities are now understood to be active modulators of host physiology, interacting with genetic, immune, metabolic, and environmental factors to influence carcinogenesis. Among these, the gut microbiome has been most extensively studied, particularly in colorectal cancer (CRC). Specific bacterial species, such as Fusobacterium nucleatum, colibactin-producing Escherichia coli, and enterotoxigenic Bacteroides fragilis, exhibit pro-tumorigenic properties[126,127]. These microbes can induce DNA damage, promote chronic inflammation, and reshape immune responses in ways that favor tumor initiation and growth. For example, F. nucleatum adheres to epithelial cells, activates oncogenic β-catenin signaling, and impairs antitumor immunity by recruiting myeloid-derived suppressor cells (MDSCs) and inhibiting natural killer (NK) cell activity[128].

Dysbiosis-driven chronic inflammation is a well-recognized driver of carcinogenesis. Microbial metabolites and ligands for pattern recognition receptors (e.g., Toll-like receptors) can amplify inflammatory responses and disrupt epithelial barriers. In inflammatory bowel disease, an established risk factor for CRC, microbial imbalances closely correlate with disease activity and barrier dysfunction, creating a tumor-permissive microenvironment.

Although direct evidence linking REG3A-modulated microbiota to cancer is currently lacking, REG3A’s known ability to shape a beneficial microbial community and preserve epithelial integrity suggests a potential protective role, particularly in colorectal and liver cancers [74]. REG3A promotes the maintenance of Firmicutes such as Lachnospiraceae and Ruminococcaceae, Gram-positive anaerobes that produce short-chain fatty acids (SCFAs) like butyrate [74]. These metabolites are generally considered protective, they reinforce epithelial barrier function, reduce inflammation, and may help suppress tumorigenesis [129] . However, their effects can be context-dependent.

Microbial influence on carcinogenesis extends beyond the gut. A well-established example is Helicobacter pylori, a major risk factor for gastric cancer, which contributes to tumorigenesis through chronic inflammation, epigenetic reprogramming, and suppression of tumor suppressor pathways, including those involving DMBT1 and E-cadherin. Notably, REG3A has been identified as a positive regulator of DMBT1, potentially counteracting H. pylori-mediated silencing of this tumor suppressor [130]. Whether this effect results from direct transcriptional regulation or is secondary to REG3A’s antimicrobial activity remains unresolved. The observed upregulation of REG3A in H. pylori-infected gastric tissue supports a possible protective role, though further research is required to elucidate its function in this context [131].

In summary, the REG3A-microbiota-cancer axis represents a promising but largely unexplored domain. While its impact on tumorigenesis remains to be definitively established, existing evidence suggests that REG3A may influence cancer risk by modulating microbial composition, inflammatory signaling, and epithelial integrity. Bridging this knowledge gap presents a valuable opportunity for future investigation.

Comment 7. Citation practices that aren't fair: The manuscript has many references, but some statements, like "REG3A acts as a metabolic modulator in multiple organs" or "REG3A modulates immune surveillance," are backed by just one source or no experimental evidence at all. On the other hand, multiple citations support specific claims, confirming the same point. Suggestion: Recheck the citation list to make sure it's balanced, especially for strong statements. Additionally, include new studies from 2022 to 2025 that have a significant impact, demonstrating that science is still advancing.

Response 7.  We thank Reviewer #2 for this constructive comment. In response, we carefully re-evaluated all citations to ensure that strong or generalized statements are adequately supported by multiple, relevant, and up-to-date references. Assertions lacking sufficient experimental backing have been clarified or reformulated with greater nuance. Additionally, we have updated the bibliography to reflect recent advances in the field, incorporating studies published between 2022 and 2025. These recent references now represent approximately 15% of the total citations.

Regarding the section on metabolism and REG3A: (i) The original title, “6. Systemic roles of REG3A: a metabolic modulator in health and disease”, has been revised to “6. Systemic roles of REG3A: Systemic Roles of REG3A: Linking Metabolism to Health and Disease”; (ii) The sentence “These systemic effects support the potential of REG3A as a metabolic modulator affecting multiple organs” has been removed;

In the section 7.2, the phrase “including mechanisms such as lectin-glycan recognition and modulation of immune surveillance” has been replaced with “including mechanisms such as lectin-glycan recognition and modulation of antitumor immune responses.”

Comment 8. The Clinical Relevance Section Needs More Work: The report briefly discusses Phase 1 and 2 studies for REG3A (p. 14) but doesn't provide detailed information on outcomes, patient groups, or clinical problems. Suggestion: Create a lengthy paragraph that discusses the current state of clinical development, including trial identifiers, therapeutic indications (such as liver failure or diabetes), and the potential for repositioning REG3A in oncology or regenerative medicine.

Response 8. In response to Reviewer 2’s request, we have added a new section entitled “8. Clinical Relevance and Therapeutic Perspectives”, which outlines the current state of clinical development for REG3A-related therapies and discusses some therapeutic opportunities. See below for the text of the new section.

  1. Clinical Relevance and Therapeutic Perspectives

Despite a robust body of preclinical evidence supporting the involvement of REG3A in diverse pathological settings, including inflammation, tissue regeneration and metabolic disorders, its translation to clinical application remains nascent. To date, only a few clinical trials have investigated REG3A or related peptides in humans. Nevertheless, early-phase studies offer encouraging translational insights and suggest the potential for broader therapeutic utility. One such example involves INGAP peptide (INGAPpp), a structural homolog of REG3A derived from hamster, which has been evaluated in two randomized, placebo-controlled Phase II clinical trials targeting diabetes mellitus. In type 1 diabetes (T1D) (NCT00071409), 63 patients received daily subcutaneous injections of 300 mg or 600 mg INGAP for 90 days [97]. The primary outcome, arginine-stimulated C-peptide via mixed-meal tolerance test, showed a significant increase in the 600 mg group compared to placebo (p = 0.0058), indicating partial β-cell functional recovery. Secondary outcomes, including HbA1c and insulin dose, improved modestly. Approximately 50% of the treatment effect persisted after a 30-day washout, suggesting transient β-cell preservation. Injection-site reactions were common (up to 90%), but no serious systemic adverse events or immunogenic responses were reported. A parallel Phase II study in type 2 diabetes (T2D), involving 126 participants, confirmed these findings [97]. The 600 mg group demonstrated significantly elevated C-peptide levels (p = 0.031) and a 0.9% reduction in HbA1c at day 90 (vs. 0.47% in placebo, p = 0.009), with durable metabolic effects persisting at follow-up (p = 0.013). Side effects were again limited primarily to injection-site reactions (about 65%), without notable systemic toxicity.

A second approach has explored recombinant human REG3A protein (ALF 5755) in the treatment of acute liver injury. A Phase IIa randomized, double-blind, placebo-controlled trial (NCT01318525) enrolled 57 adults with severe acute hepatitis (SAH) or early-stage acute liver failure (ALF) of non-acetaminophen etiology [76]. ALF 5755 (5 mg/kg, IV) was evaluated for its impact on coagulation recovery (change in pro-thrombin ratio over 72 hours). While overall results across all etiologies were neutral, a subgroup analysis in HBV and autoimmune hepatitis patients revealed a significantly improved prothrombin slope (p = 0.04) and a reduction in hospitalization duration (8 vs. 14 days, p = 0.02). Prior Phase I data confirmed that ALF 5755 was well-tolerated in healthy volunteers, with no serious adverse events, acceptable pharmacokinetics, and no immunogenicity [21].

Together, these studies provide first-in-human validation of the regenerative, an-ti-inflammatory and metabolic potential of REG3A-derived therapeutics. While clinical indications to date have focused on diabetes and ALF, REG3A's broader biological properties, including epithelial repair, preservation of intestinal barrier integrity, and modulation of innate immunity and the gut microbiota, point to its therapeutic promise in other conditions, such as mucosal injury and chronic inflammatory diseases. Its potential relevance to oncology is particularly compelling, given REG3A’s roles in immune modulation, epithelial regeneration, and microenvironmental remodeling. However, due to its context-dependent functions, including evidence of pro-tumorigenic activity in certain malignancies, careful consideration of disease context, mechanistic pathways, and target cell types is essential. Rigorous preclinical and translational studies are needed to delineate its safety profile and therapeutic window before advancing REG3A-based interventions in cancer. Moreover, recent advances in cancer immunotherapy, alongside growing evidence that gut microbiota composition influences treatment efficacy, high-light a promising opportunity for REG3A-based microbiota modulation to enhance immunotherapeutic outcomes. By reinforcing epithelial integrity and restoring a favorable microbial milieu, REG3A could synergize with immune checkpoint inhibitors to strengthen antitumor immunity and improve patient response rates.

In summary, REG3A-based interventions are emerging as promising tools in re-generative medicine, metabolic disorders, and potentially oncology, supported by early clinical safety and efficacy signals. Future studies should prioritize dose optimization, patient stratification, long-term safety evaluation, and clarification of tissue-specific effects to fully realize the therapeutic potential of REG3A in human disease.

Final Suggestion: Big Change

The work provides a thorough and promising summary of REG3A; however, it falls short in terms of critical analysis, mechanistic coherence, and translational framing in its current form. We need a major redesign that prioritizes the following areas:

  • Making mechanistic ambiguity more straightforward
  • Making translational insight stronger
  • Making graphical and tabular synthesis better
  • Getting rid of redundancies and unbalanced references

Final Response:

Round 4

Reviewer 2 Report

Comments and Suggestions for Authors

The manuscript only needs a few tweaks to be easier to read and more consistent. There is too much of the same language at first. Several words like "context-dependent," "microenvironmental complexity," and "therapeutic potential" are used, especially in Sections 7 and 8. They might be made shorter so that they are easier to read. Section 7.2 revisits how REG3A functions in hepatocellular carcinoma. The sections could be made shorter while still including all the scientific information. In Section 8, which covers clinical relevance, it would be beneficial to briefly explain why REG3A-based drugs have not made any progress in cancer treatment, despite the promising preclinical data. This information would help explain the translational gap. There are also a few minor modifications needed for the editing. Specifically, some items in Table 1 (page 5) under "Conditions" are not aligned properly. They need to be structured the same way. On page 19, the list of abbreviations should be "Colorectal Cancer" instead of "ColorRectal Cancer." The text should also show how REG3A differs from its mouse relatives, like REG3γ. This is important to avoid confusion, especially when talking about how they work in various species. These minor but crucial modifications will make the final publication more polished and accurate.